# Healthcare Complexities in Neurodegenerative Proteinopathies: A Narrative Review

**DOI:** 10.3390/healthcare13151873

**Published:** 2025-07-31

**Authors:** Seyed-Mohammad Fereshtehnejad, Johan Lökk

**Affiliations:** 1Edmond J. Safra Program in Parkinson’s Disease and Morton and Gloria Shulman Movement Disorders Clinic, Toronto Western Hospital, UHN, Toronto, ON M5T 2S8, Canada; 2Institute of Health Policy, Management and Evaluation (IHPME), Dalla Lana School of Public Health, University of Toronto, Toronto, ON M5S 1A1, Canada; 3Division of Clinical Geriatrics, Department of Neurobiology, Care Sciences and Society (NVS), Karolinska Institutet, 17177 Stockholm, Sweden; johan.lokk@regionstockholm.se

**Keywords:** neurodegenerative diseases, healthcare, pathogenesis, Alzheimer’s disease, Parkinson’s disease, frontotemporal dementia, Lewy body disease, complexity

## Abstract

**Background/Objectives**: Neurodegenerative proteinopathies, such as Alzheimer’s disease (AD), Parkinson’s disease (PD), and dementia with Lewy bodies (DLB), are increasingly prevalent worldwide mainly due to population aging. These conditions are marked by complex etiologies, overlapping pathologies, and progressive clinical decline, with significant consequences for patients, caregivers, and healthcare systems. This review aims to synthesize evidence on the healthcare complexities of major neurodegenerative proteinopathies to highlight current knowledge gaps, and to inform future care models, policies, and research directions. **Methods**: We conducted a comprehensive literature search in PubMed/MEDLINE using combinations of MeSH terms and keywords related to neurodegenerative diseases, proteinopathies, diagnosis, sex, management, treatment, caregiver burden, and healthcare delivery. Studies were included if they addressed the clinical, pathophysiological, economic, or care-related complexities of aging-related neurodegenerative proteinopathies. **Results**: Key themes identified include the following: (1) multifactorial and unclear etiologies with frequent co-pathologies; (2) long prodromal phases with emerging biomarkers; (3) lack of effective disease-modifying therapies; (4) progressive nature requiring ongoing and individualized care; (5) high caregiver burden; (6) escalating healthcare and societal costs; and (7) the critical role of multidisciplinary and multi-domain care models involving specialists, primary care, and allied health professionals. **Conclusions**: The complexity and cost of neurodegenerative proteinopathies highlight the urgent need for prevention-focused strategies, innovative care models, early interventions, and integrated policies that support patients and caregivers. Prevention through the early identification of risk factors and prodromal signs is critical. Investing in research to develop effective disease-modifying therapies and improve early detection will be essential to reducing the long-term burden of these disorders.

## 1. Introduction

Neurodegenerative proteinopathies—including Alzheimer’s disease (AD), Parkinson’s disease (PD), dementia with Lewy bodies (DLB), frontotemporal dementia (FTD), progressive supranuclear palsy (PSP), multiple system atrophy (MSA), and other related tauopathies, amyloidopathies, and synucleinopathies—represent a growing public health concern worldwide. Driven largely by population aging, the global burden of these disorders has increased significantly in recent decades and is projected to rise further in the coming years [1,2,3]. AD is the most common cause of dementia worldwide, accounting for 60–80% of cases, with more than 55 million people currently living with dementia globally and nearly 10 million new cases each year; over 60% of these cases occur in low- and middle-income countries [4]. PD is the second most prevalent neurodegenerative disorder, with its global prevalence and absolute case numbers rising significantly over recent decades, a trend attributed to increased life expectancy and demographic shifts [5]. FTD is less common, with an age-standardized global prevalence of approximately 2.3 per 100,000 among individuals aged 30–64 years, but it is a leading cause of young-onset dementia [6]. These conditions are typically characterized by the progressive accumulation of misfolded proteins in the brain, such as tau, amyloid-β (Aβ), α-synuclein, and TAR DNA-binding protein 43 (TDP-43). The subsequent neurodegenerative process results in cognitive, motor, behavioral, and functional impairments that substantially impact patients, caregivers, and healthcare systems.

Despite decades of research, the etiology of most neurodegenerative proteinopathies remains unclear and is believed to involve a complex interplay of genetic, environmental, and lifestyle factors. Adding to this complexity is the frequent presence of co-pathologies, long prodromal stages that delay diagnosis, and heterogeneity in clinical presentations ranging from cognitive decline to motor dysfunction and disease progression. While some symptomatic treatments exist, effective disease-modifying therapies (DMTs)—with meaningful clinical improvement—are still lacking, and most interventions fail to halt or reverse the underlying pathological processes. In AD, the majority of late-stage trials targeting amyloid-β (Aβ) have not translated into meaningful clinical benefit, with over 200 anti-amyloid trials failing and only limited [7], controversial approvals such as aducanumab, which was later withdrawn from the market for non-efficacy reasons [8]; tau-targeted and anti-inflammatory agents have similarly not shown clear disease modification [9]. In PD, all major DMT trials—including those of coenzyme Q10, creatine, pramipexole, and pioglitazone—have failed to slow disease progression, and ongoing studies targeting α-synuclein, mitochondrial dysfunction, and neuroinflammation have yet to yield positive results [10]. This therapeutic gap further contributes to the high burden associated with these conditions. Managing neurodegenerative diseases requires a multidisciplinary, multi-domain approach that involves collaboration among specialists, primary care providers, and allied health professionals. Such care models [11] aim to address the wide spectrum of symptoms and comorbidities associated with these disorders and to support both patients and their caregivers through the disease trajectory.

Given the increasing prevalence, clinical and pathological complexity, and high healthcare demands of neurodegenerative proteinopathies, there is a pressing need to better understand the multifaceted challenges surrounding their care. The aim of this narrative review is to explore the key healthcare complexities associated with major neurodegenerative proteinopathies, including disease etiology, progression, diagnosis, sexual dimorphism, treatment limitations, care models, caregiver burden, and economic impact.

## 2. Methods

We employed a comprehensive search strategy to identify the relevant literature for this narrative review on healthcare complexities in neurodegenerative proteinopathies. Electronic databases, including PubMed/MEDLINE, Embase, Scopus, and PsycINFO, were searched for the literature published until May 2025. We used a combination of Medical Subject Headings (MeSH) terms in MEDLINE and equivalent keywords in other databases. The search terms included variations and combinations of terms such as “neurodegenerative diseases”, “tauopathy”, “amyloidopathy”, “synucleinopathy”, “Alzheimer’s disease”, “Parkinson’s disease”, “frontotemporal dementia”, “Lewy body disease”, “progressive supranuclear palsy (PSP)”, “multiple system atrophy (MSA)”, combined with terms such as “pathogenesis”, “healthcare complexities”, “diagnosis”, “management”, “treatment”, “sex”, “ApoE4 allele”, and “caregiver burden”. Boolean operators (AND, OR) were used to combine search concepts. Additionally, hand-searching of reference lists of relevant articles and the grey literature was conducted to identify additional sources, if necessary. The search was not restricted by publication date, but focus was given to the literature published within the last decade to capture recent advancements and trends in healthcare management of these neurodegenerative conditions. The inclusion criteria encompassed articles in English that discuss the complexity of various aspects of aging-related neurodegenerative disorders related to tauopathies, amyloidopathies, and synucleinopathies.

## 3. Results

### 3.1. Multifactorial Etiology

One of the major aspects of complexity of neurodegenerative disorders is their unclear etiology. Aside from monogenic neurodegenerative diseases such as Huntington’s disease, the majority of neurodegenerative proteinopathies, particularly the commonest ones such as AD and PD, are likely multifactorial in etiology with a complex interaction between various factors.

#### 3.1.1. Alzheimer’s Disease

In the case of AD, which is characterized by a gradual decline typically in memory and cognitive function due to neuronal degeneration, symptom severity is associated with cerebral cortex protein aggregations, such as β-amyloid (Aβ) plaques, and particularly hyperphosphorylated Tau protein [12,13] leading to the formation of neurofibrillary tangles. The prevailing theory on the onset and advancement of AD, known as the amyloid cascade hypothesis, is widely embraced but still raises numerous unresolved queries [14]. The amyloid cascade hypothesis posits that AD is initiated by the accumulation of amyloid-β (Aβ) plaques, which then trigger downstream tau pathology, neurodegeneration, and ultimately cognitive decline [15]. This model is supported by genetic and biomarker studies showing that Aβ deposition often precedes tau aggregation and clinical symptoms, and it forms the basis of the Alzheimer’s Association Workgroup 2024 diagnostic criteria [16]. However, recent data indicate that only about one-third of individuals with AD pathology follow the predicted sequence of the amyloid cascade, with substantial heterogeneity in the relationship between amyloid, tau, and clinical progression, and many patients exhibiting co-pathologies or resilience to amyloid burden [16,17]. In contrast, the tau-first hypothesis suggests that tau pathology, particularly in the medial temporal lobe, may arise independently and even precede significant amyloid deposition in some individuals [17]. Tau pathology is more closely associated with neurodegeneration and cognitive impairment than amyloid, and primary age-related tauopathy (PART) can occur without significant amyloid involvement [18,19]. Some models propose that tau aggregation is a prerequisite for AD, with amyloid acting as an accelerator rather than the primary trigger [20]. The etiology of such proteinopathies in AD, however, is multifaceted and intricate. Although a minority of cases stem from dominant genetic mutations [21,22], most cases are sporadic and lack a singular genetic trigger. Various environmental and metabolic factors, including diabetes [23,24], cerebrovascular disease [25], unhealthy dietary habits during midlife [26], traumatic brain injury (TBI) [27], and chronic stress [28], are associated with heightened risk of dementia (Figure 1).

#### 3.1.2. Parkinson’s Disease

In the case of PD, the mainstay of pathogenesis is likely the aggregation of α-synuclein, another misfolded structural protein, and its propagation between the gut, brainstem, and cortical brain regions [29]. At the cellular level, complex mitochondrial and lysosomal dysfunctions are probably involved in causing neurodegeneration and neuroinflammation [29,30]. The exact mechanism of triggering the neurodegenerative process in PD is not quite clear yet; however, epidemiologic evidence strongly supports the role of environmental toxicants in this process [31]. More recently, the brain-first and body-first models of Lewy body disorders speculated that environmental risk factors trigger aggregation of alpha-synuclein either through smelling in the olfactory system or ingestion in the enteric nervous system [32].

Despite the fact that neurodegenerative diseases are highly complex and can have different etiologies, finding similarities in disease processes and pathogenesis may lead to a better understanding of the events that trigger neurodegeneration and open doors to new pan-neurodegenerative treatment options [33]. For instance, recent proteomic studies demonstrate that ApoE4 confers a systemic, pro-inflammatory immune signature across multiple neurodegenerative proteinopathies, including AD, FTD, PD, and ALS, independent of the specific proteinopathy (e.g., amyloid, tau, TDP-43, α-synuclein) [34]. This suggests ApoE4 acts as a pleiotropic immune modulator, creating a biological vulnerability to neurodegeneration rather than being AD-specific [34,35]. However, most ApoE4 carriers do not develop dementia, highlighting the importance of gene–environment and gene–disease interactions. For healthcare, this evidence supports the use of precision medicine approaches as ApoE4 status may inform risk stratification, early intervention, and biomarker development across neurodegenerative diseases. Management teams should consider the increased risk of cognitive decline, earlier onset, and potentially distinct responses to therapies in ApoE4 carriers. For caregivers, awareness of the increased risk and earlier symptom onset in ApoE4 carriers can guide anticipatory guidance, psychosocial support, and care planning [34,36].

Omics studies, including transcriptomics, proteomics, and metabolomics, are revolutionizing neurodegenerative disease diagnosis and management by identifying molecular signatures, early biomarkers, and mechanistic pathways. Genomic and transcriptomic approaches reveal disease-specific risk genes (e.g., TREM2 in Alzheimer’s disease, ATXN2 in ALS) and biomarkers, enabling more precise diagnosis and personalized medicine [37]. Proteomic and metabolomic analyses reveal deregulated pathways in AD, suggesting potential therapeutic targets like antioxidants and B vitamins [38]. Metabolomic profiling reveals sex- and genotype-specific metabolic alterations, guiding subgroup-specific therapeutic strategies [39]. Additionally, omics studies have highlighted the role of non-coding RNAs as both biomarkers and therapeutic targets in neurodegenerative diseases, expanding the landscape of potential interventions [40]. Integrated multi-omics approaches are increasingly used to identify concordant biomarkers across tissues and omics layers, improving diagnostic accuracy and enabling the development of blood-based screening tools for diseases that lack specific biomarkers [41,42]. These advances are driving a shift toward precision medicine, enabling earlier diagnosis, individualized risk assessment, and the identification of novel therapeutic targets in neurodegenerative disease management.

Disparities such as socioeconomic status, racial or ethnic bias, rural location, environmental exposures (e.g., pollution), and differences in access to healthcare systems significantly impact the healthcare, diagnosis, treatment, and management of neurodegenerative proteinopathies. Social determinants of health (SDOH) influence both disease risk and outcomes by shaping exposure to risk factors, access to early diagnosis, and availability of advanced molecular diagnostics, including omics-based approaches. For example, neighborhood-level disadvantage is associated with increased AD neuropathology, and minority and socially disadvantaged populations experience disproportionate burdens of neurodegenerative disease, in part due to limited access to specialized care and research participation [43].

### 3.2. Long Prodromal Stage

Neurodegenerative diseases are preceded by a long prodromal stage during which the pathological processes are underway, but the condition remains undiagnosed. The prodromal stage of neurodegenerative diseases represents an extended period during which the pathological processes unfold before clinical diagnosis is possible. This stage is increasingly recognized as critical for understanding disease onset and progression, as well as for developing early interventions.

#### 3.2.1. Alzheimer’s Disease

In AD, the prodromal phase can span decades, characterized by early accumulation of amyloid-beta and tau pathology in the brain [44]. Subtle cognitive decline, particularly in episodic memory, often emerges during this phase, along with non-cognitive symptoms such as depression, anxiety, and sleep disturbances [45]. As for biomarkers, in the preclinical phase, amyloid-β (Aβ) accumulation can be identified through reduced cerebrospinal fluid (CSF) Aβ42 levels or positive amyloid PET imaging years before clinical manifestations of AD [46]. The diagnostic accuracy of amyloid-beta positron emission tomography (Aβ PET) imaging for detecting the prodromal stage of AD is high for identifying cerebral amyloid pathology, with area under the curve (AUC) values of 0.92–0.93 for distinguishing mild cognitive impairment due to AD from controls; this is comparable to CSF biomarkers, and both modalities can be used interchangeably for early AD detection [47]. Synaptic dysfunction, which may even precede Aβ deposition—particularly in individuals carrying the APOE ε4 allele—can be detected using FDG-PET or functional MRI [46]. Evidence of neuronal injury appears with elevated CSF tau or phosphorylated tau, while structural MRI can reveal subtle brain atrophy associated with disease progression. These early signs and biomarker alterations provide a window of opportunity for preemptive therapeutic strategies.

#### 3.2.2. Parkinson’s Disease

Similarly, PD is preceded by a long prodromal stage—lasting at least one to two decades—characterized by a gradual loss of dopaminergic and non-dopaminergic neurons before the onset of clinical motor symptoms [48]. Clinical features during this phase often include REM sleep behavior disorder (RBD), hyposmia, constipation, and other dysautonomic symptoms, which may appear years or even decades before the hallmark motor symptoms emerge [48]. Notably, RBD is now recognized as one of the most specific prodromal markers for synucleinopathies, including PD and dementia with Lewy bodies (DLB) [49,50]. Longitudinal cohort studies demonstrate that more than 80% of individuals with idiopathic RBD will eventually develop a defined neurodegenerative syndrome, most commonly PD or DLB, with annual phenoconversion rates of approximately 6.3% and cumulative conversion rates exceeding 70% at 12 years of follow-up [51,52]. The presence of idiopathic RBD increases the lifetime risk of developing an alpha-synucleinopathy by more than 50-fold compared to the general population [52]. In fact, the association between prodromal RBD and future development of PD or DLB is so strong that recent evidence suggests idiopathic RBD should be considered an early clinical synucleinopathy syndrome [53]. These findings have highlighted the importance of identifying prodromal markers for predicting disease progression and tailoring early interventions.

#### 3.2.3. Frontotemporal Dementia

In FTD, the prodromal stage often involves changes in behavior and personality, such as apathy, disinhibition, or loss of empathy [54], reflecting early degeneration in the frontal and temporal lobes. Cognitive deficits in FTD tend to be more executive or language-based rather than memory-oriented [55]. While FTD shares overlapping features with other neurodegenerative diseases, such as mood disturbances and subtle behavioral changes, its distinct prodromal profile underscores the heterogeneity of neurodegenerative conditions.

Despite these advances in understanding, the duration of the prodromal stage remains highly variable, influenced by genetic, environmental, and lifestyle factors, as well as the arbitrary criteria used to define disease onset. This variability reflects not only differences in disease biology but also the limitations of current diagnostic frameworks. Importantly, many symptoms in the prodromal phase, whether motor, cognitive, or behavioral, are already present but remain undetected due to the insensitivity of existing clinical tools [56]. Emerging research emphasizes the need for improved biomarkers and advanced imaging techniques to identify early changes in brain structure and function, as well as digital health tools to monitor subtle symptomatology. Understanding the prodromal stage of neurodegenerative diseases is essential for advancing early diagnosis and prevention. It represents a critical time window where interventions targeting modifiable risk factors, such as physical inactivity, poor sleep, and cardiovascular health, could significantly alter the trajectory of disease progression. Moreover, increased awareness of this stage could lead to the development of tailored prevention strategies and novel therapies, shifting the focus from reactive care to proactive management of neurodegenerative diseases.

### 3.3. Co-Pathologies

An emerging complexity in the field of neurodegenerative proteinopathies is the high prevalence of co-pathologies, particularly in AD and related dementias. Rather than being driven by a single misfolded protein, many neurodegenerative diseases involve the aggregation of multiple proteinopathies, including amyloid-β (Aβ), phosphorylated tau (p-tau), α-synuclein, and TDP-43 [57]. These co-existing pathological entities are not merely incidental findings; they likely influence the clinical phenotype, progression, and response to therapy. For AD, it is proposed to shift from a phenotype-based classification of neurodegenerative diseases to a molecular biomarker-based framework, encapsulated in the AT(N) system, which identifies the presence of amyloid (A), tau (T), and neurodegeneration (N) [58]. However, even this model may underestimate the true complexity of disease pathology, as many patients—especially those with late-onset AD—harbor additional pathologies such as TDP-43 inclusions or vascular lesions that contribute to clinical heterogeneity.

Supporting this, a large autopsy study of 1647 individuals demonstrated that the majority of patients with neurodegenerative diseases exhibit multiple concurrent pathologies, many of which extend beyond age-associated changes [59]. The study identified 161 unique pathological combinations, with the most frequent scenario in AD patients being the co-occurrence of Aβ, tau, cerebral amyloid angiopathy (CAA), and TDP-43 [59]. Notably, in the clinical Alzheimer’s cohort, more than half of the cases had five or more pathologies, often linked to older age, longer disease duration, and APOE ε4 status [59]. For instance, TDP-43 pathology, present in over half of older adults with Alzheimer’s pathology, is associated with faster cognitive decline, greater hippocampal atrophy, and more severe neurodegeneration [60]. Its spread beyond the amygdala significantly increases dementia risk and progression, with a synergistic effect that worsens tau aggregation, neuronal loss, and cognitive symptoms [60,61]. These findings challenge traditional clinicopathological correlations and highlight the need for multimodal diagnostic approaches that capture this molecular and pathological diversity. Co-pathologies are not only common in neurodegenerative diseases but also play a central role in their underlying pathophysiology. Their presence has significant implications for the development of biomarkers, the design of clinical trials, and the advancement of future therapeutic strategies.

The ATX(N) framework expands upon the traditional AT(N) biomarker model by incorporating additional biomarkers (X) that capture non-amyloid, non-tau pathologies, such as TDP-43, α-synuclein, neuroinflammation, and vascular injury, thereby better reflecting the biological heterogeneity of Alzheimer’s disease [62]. This approach allows for a more nuanced stratification of patients, improves the prediction of clinical outcomes, and supports the design of more targeted clinical trials by accounting for the frequent and clinically relevant co-pathologies that drive disease progression and therapeutic response [62,63].

### 3.4. Progressive Nature

Another common feature of neurodegenerative disorders, such as AD, PD and atypical parkinsonism, FTD, and other dementias, is their gradually progressive nature, marked by the relentless deterioration of neuronal structures and functions over time. This progression is driven by pathological propagation, wherein abnormal proteins such as amyloid-beta, tau, alpha-synuclein, or TDP-43 misfold and aggregate, spreading in a prion-like manner across neural networks [64]. These pathological processes often follow distinct anatomical trajectories, correlating with the clinical phenotypes and the worsening of motor, cognitive, or behavioral symptoms [65,66]. Despite the commonality of progressive decline across neurodegenerative disorders, the rate of progression varies significantly between different disease entities and even among individuals with the same predominant pathology; for instance, cognitive decline tends to progress more rapidly in FTD, corticobasal syndrome CBS, and DLB compared to typical AD [67], while motor decline is notably faster in progressive supranuclear palsy PSP and MSA compared to typical PD [68]. Published evidence demonstrates that the annual decline in Mini-Mental State Examination (MMSE) scores is faster in patients with FTD than in those with AD. Specifically, one study found that the mean annualized MMSE decline was 4.7 points per year in FTD compared to 3.2 points per year in AD [69]. Even in patients all with idiopathic PD, various subtypes have been defined to explain its heterogeneous progression. When compared to individuals who manifest with a motor-dominant PD and have the least burden of non-motor features at baseline, the so-called ‘diffuse malignant’ subtype of PD develops dementia faster and has an overall rapid course of progression [70,71,72].

### 3.5. Sex Differences

Biological sex has a significant impact on the onset, progression, and healthcare outcomes of neurodegenerative proteinopathies. Women have a higher prevalence and lifetime risk of AD, experience faster cognitive decline, and show greater amyloid and tau pathology burden compared to men, even after adjusting for longevity [73,74]. In contrast, PD is more prevalent in men, but women with PD often have later onset, more rapid motor progression, greater risk of levodopa-induced dyskinesias, and more prominent mood disorders [75,76]. Sex differences in immune response, neuroinflammation, and hormonal regulation contribute to these disparities, with recent data showing that microglial activation mediates amyloid-to-tau pathology more strongly in females, potentially accelerating disease progression and symptom burden in women with AD [77,78]. Another aspect of biological sex that impacts on the pathogenesis of neurodegenerative diseases is the gut microbiome. Females exhibit distinct gut microbiome profiles compared to males, driven by sex hormones (notably estrogen), genetic factors, and age-related changes, particularly after reproductive senescence [79]. These differences modulate neuroinflammatory responses, amyloid and tau pathology, and microglial activation, contributing to the higher prevalence and faster progression of Alzheimer’s disease in women [79,80]. In PD, sex-specific gut dysbiosis patterns have been identified, with male and female patients showing different dominant microbial taxa and associated metabolic pathways, which correlate with distinct brain functional changes and symptom profiles [81].

Healthcare outcomes and access to multidisciplinary care are also affected by sex and gender. Women with AD are less likely to receive timely diagnosis, have higher rates of institutionalization, and face greater caregiver burden, while men with PD may have less access to allied health services and support [82]. Sex bias persists in clinical trial enrollment, with underrepresentation of women in PD trials and men in AD trials, limiting the generalizability of failed antibody-based clinical trial results and impeding the development of sex-specific therapies [83]. Multidisciplinary care models, though effective, face challenges in addressing these disparities due to funding limitations, workforce shortages, and lack of sex- and gender-sensitive protocols, particularly in real-world and resource-limited settings. Addressing these gaps requires intentional integration of sex and gender considerations into care pathways, research design, and health policy to achieve equitable and personalized management of neurodegenerative proteinopathies [82,84].

### 3.6. Lack of Efficient Disease-Modifying Therapies

There is currently no preventative therapy or intervention capable of halting or reversing the pathological cascade in neurodegenerative diseases of aging [85]. Most available treatments focus on symptomatic relief, leaving the underlying disease mechanisms unchecked. Despite decades of research, effective disease-modifying therapies for neurodegenerative proteinopathies remain elusive. One of the central challenges lies in the biological complexity of these disorders. Neurodegenerative diseases such as AD, PD, DLB, and FTD rarely follow a singular pathogenic trajectory. Instead, they often involve overlapping molecular cascades, co-pathologies (e.g., concurrent amyloid, tau, TDP-43, or α-synuclein deposits), and prominent interactions with age-related changes. This convergence of multiple pathologies in a single individual complicates both diagnosis and therapeutic directing. Thus, targeting a single protein (i.e., amyloid) will be unlikely to solve the entire problem of these entities. Adding to this complexity is the incomplete understanding of the causal role played by the hallmark misfolded proteins. While the presence of α-synuclein, amyloid-β, or tau aggregates is characteristic of specific diseases, their precise role in initiating or driving neurodegeneration is still debated [86,87]. This uncertainty has direct implications for therapeutic design: interventions targeting a single protein species may have limited impact in the presence of multiple co-existing pathological processes with yet unclear roles in pathogenesis. Aging, the greatest risk factor for most neurodegenerative diseases, further confounds therapeutic efforts by introducing a range of cellular and systemic changes [88] that render the nervous system more vulnerable and less responsive to treatment.

#### 3.6.1. Alzheimer’s Disease

The most extensively studied target—amyloid-β in AD—therapeutic breakthroughs have been modest [7]. Recently approved amyloid-targeting agents, such as monoclonal antibodies—Aducanumab and Lecanemab—have demonstrated statistically significant effects on biomarker reduction and marginal slowing of cognitive decline [89]. However, the clinical benefits appear to be slight and of uncertain meaningfulness for patients’ daily functioning [7]. Furthermore, these therapies are associated with complications, including amyloid-related imaging abnormalities (ARIAs), which may lead to cerebral edema or hemorrhage [90].

#### 3.6.2. Parkinson’s Disease

In the case of PD, phase 2 trials of prasinezumab [91] and cinpanemab [92], both targeting aggregated α-synuclein, have failed in clinical trials. In the PASADENA trial, prasinezumab did not significantly improve the primary endpoint compared to placebo, and infusion reactions were more common in active treatment groups [91]. The trial concluded that targeting extracellular α-synuclein alone was insufficient to slow disease progression.

The debate surrounding their utility underscores the need for a more nuanced and integrative understanding of neurodegenerative disease mechanisms to develop effective and safe treatments. This lack of efficient disease-modifying therapies emphasizes the urgent need for deeper understanding of molecular pathways, improved biomarkers for early detection, and novel therapeutic approaches [93].

### 3.7. Multi-Domain Multidisciplinary Care

A comprehensive care approach is required to address the diverse and complex needs of individuals with neurodegenerative diseases such as AD and PD. These conditions often present a constellation of symptoms affecting not only cognitive functions or motor skills, but also mood and behavior, speech, sleep pattern, swallowing capacity, and overall quality of life. Therefore, a team-based care model involving various healthcare professionals is crucial for optimizing patient outcomes [94]. Various clinical trials and observational studies have demonstrated that this multidisciplinary care model, sometimes known as collaborative care, significantly improves the quality of care and, in particular, the behavioral and psychological symptoms of patients with progressive neurodegenerative diseases like Alzheimer’s dementia [95,96] and PD [97,98,99].

Apart from neurologists, geriatricians, psychiatrists, and primary care physicians (i.e., family physicians), allied health team members play a pivotal role in making a multidisciplinary care model possible for patients with neurodegenerative disorders (Table 1).

Physiotherapists are integral members of this multidisciplinary team, focusing on improving physical function and mobility. They assess gait, balance, and strength deficits and design tailored exercise programs to maintain or improve mobility and prevent falls. For instance, in PD and atypical parkinsonism syndromes, physiotherapists may employ specific exercises targeting flexibility and coordination to help manage motor symptoms like tremors, axial rigidity, and freezing of gait [100]. Occupational therapists play a fundamental role in enabling patients to perform daily activities independently and ensuring home safety given the increased risk of falls. They evaluate the patient’s ability to engage in self-care tasks such as dressing, grooming, and meal preparation. As part of the multidisciplinary care model, occupational therapists recommend adaptive equipment and modifications to the home environment to enhance safety and promote functional independence, such as using grab bars in bathrooms or installing ramps for wheelchair accessibility [101]. Speech–language therapists specialize in addressing communication and swallowing difficulties commonly seen in neurodegenerative diseases. The spectrum of swallowing impairment varies in neurodegenerative disorders, from motor oropharyngeal dysphagia in PD [102] to oromandibular dystonia in atypical parkinsonism [103] and oro-buccal apraxia in CBS [104]. Likewise, communication problems occur due to a wide range of etiologies, from aphasia in primary progressive aphasia [105] to severe hypophonia or adynamic speech in PSP [106], all of which need to be assessed by an experienced speech–language therapist. Speech–language therapists provide strategies to improve communication skills, such as using alternative communication devices or techniques (e.g., Lee Silverman voice treatment (LSVT) in PD) [107]. They also work on swallowing exercises and modifications to prevent aspiration and ensure safe eating and drinking. Social workers play a vital role in addressing the psychosocial aspects of neurodegenerative diseases. They provide emotional support to patients and families, facilitate access to community resources and support groups, and assist with navigating complex healthcare systems [108]. Social workers advocate for patients’ rights and help address financial and logistical challenges that may arise due to the disease.

In addition to the multidisciplinary allied health team, the role of primary care physicians, particularly family physicians, is vital in the holistic management of neurodegenerative diseases. Family physicians serve as the first point of contact for patients and play a key role in coordinating care across various specialties. They provide comprehensive assessments, monitor disease progression, and manage comorbidities such as hypertension, diabetes, and cardiovascular diseases that often accompany neurodegenerative conditions such as vascular dementia. In the case of PD as an example, family physicians can manage autonomic symptoms like orthostatic hypotension and constipation, treat neuropsychiatric symptoms like depression and sleep disorders, and help recognize and treat psychosis in advanced stages [109]. Furthermore, family physicians offer ongoing support, education, and counseling to patients and their families, helping them navigate the complexities of the disease and access appropriate resources. In most countries with a referral healthcare system, patients with neurodegenerative disorders visit their primary care physicians more often than specialists. A dynamic communication between the general practitioner and the specialists (i.e., movement disorders specialist neurologist, cognitive neurologist, geriatrician) can significantly benefit patients’ quality of care and life [110].

In some neurodegenerative diseases like PD, specialist nurse practitioners with advanced training in advanced therapies such as levodopa–carbidopa intestinal gel (DuoDopa) and deep brain stimulation (DBS) contribute significantly to patient care [111]. These nurse practitioners work closely with movement disorders neurologists and other healthcare professionals to optimize treatment outcomes. For example, in patients with advanced PD, nurse practitioners proficient in managing DuoDopa therapy can provide continuous infusion therapy, closely monitor response, and adjust medication doses as needed to minimize motor fluctuations and improve quality of life. Similarly, nurse practitioners specializing in DBS assist in patient selection, preoperative assessments, postoperative care, and programming of stimulator settings, ensuring optimal therapeutic outcomes and patient safety [112].

The collaborative efforts of these healthcare professionals, along with neurologists, dietitians, and psychologists, create a holistic care approach tailored to the unique needs of each patient. This team-based care model not only improves symptom management but also enhances overall quality of life by addressing the physical, cognitive, emotional, and social aspects of neurodegenerative diseases. In randomized controlled trials, the collaborative care approach to dementia management significantly reduced behavioral and psychological symptoms, as measured by a 7.45-point decline on the Neuropsychiatric Inventory (NPI), compared to usual care [113]. This model also reduced caregiver burden and improved quality of life for both patients and caregivers [113]. These effects are most pronounced in community-dwelling populations and are associated with delayed institutionalization and improved health-related quality of life [114]. Telehealth interventions are feasible and increasingly implemented for the care of patients with AD and PD even in low- and middle-income countries (LMICs), with evidence supporting their effectiveness, accessibility, and unique challenges in these settings [115,116]. Randomized trials and meta-analyses demonstrate that telehealth, including video consultations, telerehabilitation, and remote cognitive training, can significantly improve global cognitive function, memory, motor and non-motor symptoms, and quality of life in both AD and PD, with effect sizes comparable to in-person care for many outcomes [117,118]. High rates of patient and caregiver satisfaction, reduced travel time and costs, and improved access to specialist care, especially in rural or underserved regions, are consistently reported [119].

Implementing multidisciplinary care models for neurodegenerative diseases like AD and PD faces challenges like funding constraints, care coordination complexity, patient access barriers, and integration into existing health systems. Funding is a major barrier, while workforce shortages and effective communication across disciplines and care settings are logistically demanding [120]. Access to healthcare is hindered by geographic disparities, digital literacy, and technology infrastructure, especially in rural areas and low- and middle-income countries [118,121]. Telehealth and community-based models can help, but integration requires system-level buy-in and ongoing education [122]. Overall, while multidisciplinary and telehealth-enabled care models are feasible and can reduce costs and improve outcomes, their real-world implementation demands solutions for sustainable funding, workforce development, care coordination infrastructure, and health system integration, tailored to local resource environments.

### 3.8. Caregiver Burden

Neurodegenerative proteinopathies, such as AD, PD, and FTD, place a profound burden on caregivers due to the progressive and multi-domain dysfunctions associated with these disorders. Mobility difficulties, cognitive decline, mood, and behavioral problems are common features that significantly increase caregiving demands [123,124]. Patients with motor impairments, such as rigidity, bradykinesia, and gait disturbances, often require substantial physical assistance, which can lead to caregiver fatigue and even physical injury [125]. Similarly, cognitive decline—ranging from forgetfulness to severe limb apraxia—necessitates constant supervision and support with daily activities, adding emotional and time pressures. Mood and behavioral issues, including apathy, depression, irritability, aggression, and wandering, pose unique challenges, further escalating the stress experienced by caregivers. A systematic review demonstrated that depression, aggression, and sleep disturbances were the most commonly identified patient symptoms contributing to caregiver burden, though a broad spectrum of symptoms was also associated with caregiver stress and depression [126].

Family members of individuals with neurodegenerative disorders who assume caregiving roles are particularly vulnerable to burnout, a state of physical, emotional, and mental exhaustion resulting from prolonged stress. The emotional strain of witnessing a loved one’s decline, coupled with the isolation caused by caregiving responsibilities, can lead to anxiety and depression. Approximately 40% of dementia caregivers experience symptoms of depression [127]. As revealed by another systematic review, burnout syndrome in caregivers of patients with dementia—in a vicious cycle—negatively impacts their quality of life and is linked to patient depressive and anxious symptoms as well as abusive behavior by caregivers [128]. Financial pressures, including the costs of long-term care and potential loss of income, compound the burden, while caregivers’ own health often deteriorates due to neglect of their well-being [129]. Addressing this burden requires a multifaceted approach that includes education, respite care, psychological support, policy interventions, and comprehensive insurance schemes [129]. For example, the REACH-II (resources for enhancing Alzheimer’s caregiver health) intervention was delivered as a six-month, multicomponent program involving in-home and telephone sessions targeting caregiver self-care, safety, emotional well-being, social support, and management of care recipient behaviors [130]. It significantly improved caregivers’ self-reported health, sleep, and emotional well-being, while reducing burden and distress across diverse racial and ethnic groups—effects largely mediated by reductions in depressive symptoms [130]. Additionally, providing caregivers with resources to manage disease progression, offering in-person and virtual educational platforms, and implementing financial and workplace flexibility programs can significantly alleviate stress [131,132]. Supporting caregivers in these ways not only improves their quality of life but also enhances the care they provide for patients.

### 3.9. Costly Management

Neurodegenerative diseases such as PD, AD, and other dementias are characterized by their gradual progression, leading to substantial and escalating costs over time. Direct costs encompass medical expenses such as ongoing diagnostics, pharmacological treatments, referrals to emergency departments and hospitalizations due to disease complications (i.e., recurrent falls, urinary tract infection, pneumonia, failure to cope), specialized outpatient care, and the utilization of long-term care services [133,134]. Indirect costs, however, often outweigh the direct expenditures, including lost productivity, informal caregiving by family members, and diminished quality of life for both patients and caregivers [133]. Caregivers face financial challenges as they often need to reduce or abandon their own professional commitments to provide care, further compounding the economic burden [135]. The costs associated with neurodegenerative diseases increase dramatically in advanced stages, when patients often require continuous care, including home healthcare services, nursing facilities, or hospitalizations for complications such as falls or infections. A study comparing end-of-life costs found that individuals with neurodegenerative diseases, including AD, PD, and ALS, had higher emergency department utilization compared to those with malignant brain tumors, reflecting the greater healthcare demands of these chronic conditions in their final stages [136]. Advanced supportive technologies, multidisciplinary teams, and individualized care plans add further financial strain to families and healthcare systems.

#### 3.9.1. Alzheimer’s Disease

A recent report estimated that in 2019, the global direct healthcare spending on AD and related dementias totaled USD 260.6 billion, with informal care costs reaching USD 354.1 billion, accounting for 57% of the total care costs [137]. By 2050, direct healthcare spending on dementias is projected to rise to USD 1.6 trillion, representing 9.4% of worldwide health expenditures, while informal care costs are expected to reach USD 0.9 trillion [137]. In line with the previous discussion, community-based day programs and multidisciplinary dementia care models, including those delivered via telehealth, have demonstrated cost savings for patients with AD by reducing emergency department visits, delaying institutionalization, and lowering overall healthcare expenditures. Randomized clinical trials of collaborative care models such as the Care Ecosystem show a mean monthly reduction in total Medicare costs of USD 526 per patient, with cumulative cost savings of USD 3290 in the first 6 months and USD 3027 in the subsequent 6 months compared to usual care, primarily through decreased acute care utilization and improved care coordination [120]. These programs also improve patient quality of life and reduce caregiver burden and depression, which are associated with further downstream cost savings [138].

#### 3.9.2. Parkinson’s Disease

In the case of PD, patients frequently experience a significant loss of productivity during their most economically active years, typically between the ages of 50 and 70 [139]. A systematic review found that individuals with PD retire 4–7 years earlier than the general population, with 23–75% attributing their early retirement to PD [140]. This is primarily due to physical limitations caused by motor symptoms, such as bradykinesia and tremors, as well as non-motor symptoms like fatigue, cognitive decline, and mood disorders. These impairments often lead to early retirement, reduced work hours, or a complete inability to maintain employment. Similar to other neurodegenerative disorders, multidisciplinary care models may also help reduce the cost of care for PD. Specialist nursing services in community settings have been associated with significant reductions in hospital length of stay (by 0.37–0.76 days per admission) and net cost savings of up to USD 8600 per person with PD over three years, primarily by preventing avoidable admissions and improving care coordination [141].

## 4. Conclusions

While this review offers a broad synthesis of current understanding, it is limited by its English-only scope and narrative methodology, which lacks the systematic rigor of formal review protocols. Nonetheless, this review aims to provide a comprehensive framework to inform policy, research, and care innovation in this increasingly complex area. Neurodegenerative proteinopathies represent some of the most complex and costly challenges facing modern healthcare systems. The complexities highlighted in this review—particularly the multifactorial interplay of biological, clinical, and social factors, the extended prodromal phases, and the high prevalence of co-pathologies—fundamentally challenge traditional paradigms of diagnosis, treatment, and care delivery in neurodegenerative proteinopathies. These conditions often defy linear diagnostic pathways due to overlapping symptoms and evolving pathology over time. The prolonged prodrome complicates early detection and timely intervention, while co-existing pathologies obscure clinical presentation and reduce the efficacy of single-target treatments. As such, traditional models centered on late-stage diagnosis and single-pathology interventions are insufficient. In addition, there are fundamental challenges in current healthcare systems. Early identification is often hindered by the absence of clear biomarkers or infrastructure to support timely screening in primary care. The dynamic and unpredictable disease trajectory complicates resource planning, requiring flexible models that can adapt to shifting levels of patient need. As demonstrated in this review, the burden of these diseases extends far beyond the individual patient, profoundly affecting caregivers, healthcare infrastructure, and the economy. Prevention is key. Addressing modifiable risk factors, identifying individuals in the prodromal phase, and implementing early, personalized interventions may significantly alter the trajectory of these diseases. Moreover, these growing costs underscore the urgent need for innovative and scalable care models, proactive health policies that offer financial and psychosocial support to patients and caregivers, and sustained investment in research to discover effective therapies. A forward-looking approach that prioritizes early detection, prevention, and interdisciplinary care will be critical in mitigating the long-term human and economic impact of neurodegenerative proteinopathies. Next steps should include piloting prodromal-stage screening programs using emerging biomarkers, advancing the development of disease-modifying therapies that address the multifactorial and complex pathogenesis, and expanding multidisciplinary care models to deliver individualized, coordinated support.

## Figures and Tables

**Figure 1 healthcare-13-01873-f001:**
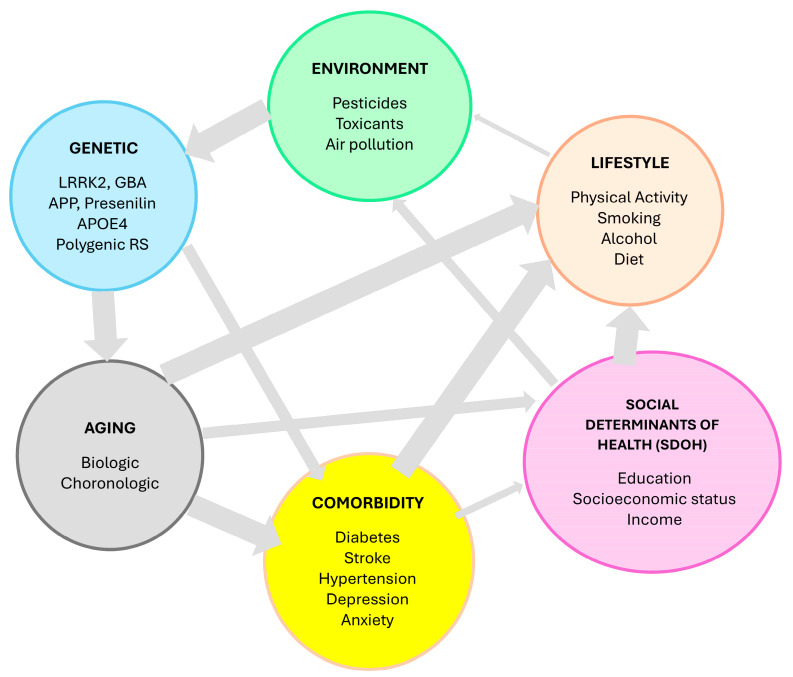
The complex interplay between various risk factors for neurodegenerative disorders such as Alzheimer’s disease and Parkinson’s disease.

**Table 1 healthcare-13-01873-t001:** Roles of multidisciplinary team members in neurodegenerative disease care.

Team Member	Primary Roles and Contributions
Neurologist	Diagnosis and management of neurodegenerative disorders; coordination of specialist care (e.g., movement disorders, cognitive neurology).
Geriatrician	Comprehensive care for older adults with multimorbidity; functional and cognitive assessments.
Psychiatrist	Management of mood, behavioral, and psychotic symptoms; support for psychiatric comorbidities.
Primary Care Physician	First point of contact; chronic disease management, comorbidities (e.g., hypertension, diabetes); care coordination and continuity; patient and caregiver education.
Physiotherapist	Assessment and treatment of mobility, gait, balance, and strength; fall prevention; exercise programs tailored to motor symptoms (e.g., tremor, rigidity, freezing of gait).
Occupational Therapist	Support for daily living activities; home safety assessments; adaptive equipment recommendations; environmental modifications.
Speech–Language Therapist	Assessment and management of communication and swallowing impairments; swallowing exercises; speech therapy (e.g., LSVT); use of augmentative communication devices.
Social Worker	Emotional support; resource navigation; financial and legal advocacy; connection to community services and caregiver support.
Specialist Nurse Practitioner	Expertise in advanced therapies (e.g., DuoDopa, DBS); monitoring and adjusting therapy; patient education; support across treatment continuum.
Dietitian	Nutritional assessments; diet modifications for swallowing difficulties or metabolic needs; support for healthy aging.
Psychologist	Cognitive and emotional assessments; psychotherapy for depression, anxiety, and caregiver stress; behavioral interventions.

## Data Availability

Not applicable.

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
