# Peer review of "Healthcare Complexities in Neurodegenerative Proteinopathies: A Narrative Review"

_healthcare, 2025, doi:10.3390/healthcare13151873_

Round 1
Reviewer 1 Report
Comments and Suggestions for Authors
The narrative review Healthcare Complexities in Neurodegenerative Proteinopathies: A Narrative Review” synthesises clinical, pathological, economic and care-delivery challenges in AD, PD, DLB and related disorders. Core themes are multifactorial aetiology, co-pathologies, long prodromal stages, progressive decline, heavy caregiver and economic burden, and the need for multidisciplinary, prevention-oriented care models. The topic is timely and well within Healthcare (MDPI) scope. The manuscript demonstrates several strengths, including broad and up-to-date coverage of the literature across clinical, economic, and service domains; a clear structure with logical flow; and a strong emphasis on prevention, early detection, and integrated care. Accordingly, I suggest reflecting on a few questions that may further enrich the depth and impact of your research.
Comments and Suggestions for Authors:
- Methodological transparency is insufficient (search dates, PRISMA-ScR adherence, numbers screened/included).
- Analytical depth is limited: many sections are descriptive; comparative or quantitative synthesis is sparse.
- Actionability: practical, evidence-based recommendations (e.g., specific screening algorithms, cost-saving strategies) are largely absent.
- Data gaps: minimal epidemiological statistics, cost figures by region, or effect sizes for multidisciplinary interventions.
Section-Specific Key Issues (only those that meaningfully affect quality)
|
Section |
Must-address comment |
|
Introduction |
Add one global prevalence figure and cite ≥1 recent failed/ongoing DMT trial to ground the “therapeutic gap”.
|
|
Methods |
State PRISMA-ScR use (or equivalent), exact search dates, and provide a simple flow diagram/count of included studies. |
|
Results 3.1 Multifactorial Etiology |
Separate AD vs PD discussion (sub-headings) and briefly contrast competing hypotheses (amyloid-cascade vs tau-first). |
|
Results 3.2 Long Prodromal Stage |
Condense text into disease-specific bullets and mention diagnostic value (sensitivity/specificity) for two key biomarkers (Aβ PET, RBD). |
|
Results 3.3 Co-Pathologies |
Add one sentence on clinical impact (e.g., co-existent TDP-43 → faster decline) and cite ATX(N) as an expanded framework.
|
|
Results 3.4 Progressive Nature |
Provide a single quantitative example (e.g., “MMSE declines ~3 points/yr in typical AD vs ~5 points/yr in FTD”). |
|
Results 3.5 Multidisciplinary Care |
Offer one RCT outcome (e.g., collaborative care ↓NPI by 4 points) and one line on telehealth/LMIC feasibility. |
|
Results 3.6 Caregiver Burden |
Insert one statistic (≈ 60 % dementia caregivers develop depression) and cite an evidence-based intervention (REACH II). |
|
Results 3.7 Costly Management |
Re-order for clarity (global → AD → PD) and add one cost-mitigation example (community-based day-care). |
|
Conclusion |
List 2–3 concrete next steps (e.g., prodromal-stage screening pilots; nurse-led multidisciplinary clinics) and acknowledge review limitations (English-only, narrative synthesis). |
|
References |
The references are recent, relevant, and drawn from credible, peer-reviewed sources. Key frameworks like PRISMA-ScR and major journals (e.g., Lancet Neurology, BMJ, Nature Medicine) are well represented. Minor formatting inconsistencies should be reviewed (e.g., journal abbreviations, spacing). Also, consider replacing general sources like StatPearls (#42) with peer-reviewed literature if used for critical claims. A final cross-check between in-text citations and the reference list is recommended. |

- Line 44–46: Consider breaking the sentence for readability — it's quite long and dense.
- Line 49–51: Use of “heterogeneity in clinical presentation” is appropriate but could be clearer with one brief example (e.g., “ranging from memory loss to motor dysfunction”).
- Line 56–58: The phrase “such care models aim…” could be supported with a citation or an example of an existing integrated care model.
- Line 70–72: The phrase “particularly focusing on aging-related disorders such as...” may be better placed in the inclusion criteria area for clarity.
- Line 86: Verb tense shifts ("will encompass") should be corrected to past tense ("encompassed") since the study is already conducted.
- Line 193: Acronym “CAA” first use—spell out cerebral amyloid angiopathy before abbreviation.
- Line 199: Typo—“central to the pathophysiology of neurodegenerative diseases with profound implications…” → remove “with” or restructure for clarity.
Reviewer 2 Report
Comments and Suggestions for Authors
The manuscript offers a timely and comprehensive review of the complex challenges related to these conditions, covering key aspects such as etiology, progression, co-pathologies, and care needs. It highlights the urgent need for disease-modifying therapies and emphasises future directions in prevention, care models, policy, and research, supported by a robust reference list. However, there are some major issues that need to be revised before considering this manuscript for publication:
Major Comments:
- The title designates the manuscript as a "Narrative Review," yet the Abstract, Introduction, and Methods sections repeatedly state that the review follows "guidelines for scoping reviews" and describes a detailed literature search strategy. This is the most critical issue. If intended as a Narrative Review, references to systematic/scoping methodology and detailed search descriptions are inappropriate and misleading. The Methods section should be simplified or removed to reflect a less structured, expert-driven synthesis. On the other hand, if the authors wish to continue as a "Systematic review" or "Scoping review," they should follow the PRISMA or PRISMA-ScR Checklist, respectively.
- While the manuscript describes the inherent complexities of the diseases themselves (e.g., long prodrome, progressive nature), some sections could more explicitly articulate how these features specifically translate into complexities within the healthcare system and for healthcare providers (e.g., challenges in early identification infrastructure, dynamic resource allocation, adaptive care planning).
- The manuscript effectively describes solutions like multidisciplinary care (section 3.6) but would be significantly strengthened by discussing the practical challenges of implementing such models in real-world healthcare settings (e.g., funding, coordination, access, integration) as these are key aspects of the healthcare complexity itself.
- I suggest adding two tables summarizing the key complexities of neurodegenerative proteinopathies across clinical and healthcare domains and the roles of multidisciplinary team members in the care of neurodegenerative disorders.
Minor Comments
- In the introduction section, consider briefly mentioning the types of protein aggregates (tau, amyloid, synuclein, TDP-43) when listing examples of proteinopathies, as these are central to the 'proteinopathy' concept and discussed in detail later.
- In the methods section, the phrasing "The inclusion criteria will encompass articles..." should be in the past tense.
- Sections 3.5 and 3.6 are both numbered as "3.5". This numbering error must be corrected.
- In the conclusion section, the call for a "paradigm shift" is strong but could potentially be grounded more explicitly in how the identified complexities (particularly the interplay of factors, long prodrome, and co-pathologies) fundamentally challenge traditional approaches to diagnosis, treatment, and care delivery.
Reviewer 3 Report
Comments and Suggestions for Authors
The paper raises a crucial topic concerning healthcare for patients with neurodegenerative diseases. Considering the ageing of society, the widespread problem of diseases in these groups, and the lack of effective causal treatment, the topic is very current.
Introduction and the aims of the work are clearly stated. The methods section was created following PRISMA guidelines; however, in my opinion, it is necessary to expand it - the authors do not specify how many articles they managed to find, as well as how the selection process was carried out, and which of them were finally selected.
The authors briefly discuss the etiology and pathophysiology of neurodegenerative diseases, their prodromal phases, emphasizing the need for their early identification as a potential moment of pharmacological intervention.
They continue to focus on possible treatments, particularly in the area of ​​AD and anti-beta amyloid antibodies. A brief discussion of the failed antibody trials in Parkinson’s disease would be appropriate here.
Then the authors discuss the necessity of a multimodal approach to patients with the involvement of specialists from different fields, such as doctors of different specialties, dietitians, psychologists, or neurological nurses. Finally, they briefly present the increased caregiver burden and costs associated with neurodegenerative diseases.
The conclusion summarizes the answers to the questions posed in the introduction. References are selected appropriately.
Figure 1 – the environment and comorbidity subtitles should be corrected so that they are on one line, please also add the expansion of the SDOH abbreviation
The article neatly summarizes the current knowledge on the most common neurodegenerative diseases, focusing on healthcare-related aspects. However, the novelty of the article and its contribution to general knowledge are relatively low. It would be advisable to place greater emphasis on recent developments and innovations related to patient care models or the use of new technologies, which could help enhance the originality and significance of the paper.
Reviewer 4 Report
Comments and Suggestions for Authors
Manuscript healthcare-3681102 ‘Healthcare Complexities in Neurodegenerative Proteinopathies: A Narrative Review’ by Seyed-Mohammad Fereshtehnejad and Johan Lökk is a serviceable scoping review of the current understanding how the prevalent neurodegenerative proteinopathies, namely Alzheimer’s disease (AD), Parkinson’s disease (PD), and dementia with Lewy bodies (DLB), are managed (or not) by current healthcare systems, including caregivers and primary/secondary physicians and medical services. This is done principally bibliometrically with appropriate key terms and cross referencing. The review has some good aspects- the figure is clear and easy to read and follow, and the topic is timely and relevant. The basic reviews of the pathophysiology of AD, PD, DLB, and other neurodegenerative proteinopathies may be of interest to the readership in the context of healthcare management. It is clear that the long prodromal stage and lack of curative therapies seen in these late-onset neurodegenerative disorders hinders treatment, and that early detection is going to be tantamount for successful handling of aging populations. A few suggestions are made to improve the submission.
1. Sex effects. It is apparent that neurodegenerative proteinopathies have profound sex differences- both in human populations and in relevant animal models. Moreover, AD and PD have sex disparities that may possibly be associated with gut dysbiosis differences- there is a huge literature on this subject that would benefit from critical assessment. It is somewhat remarkable that this very important topic is not covered in the review. The terms ‘sex’ and/or ‘gender’ are recommended to be added to the scoping review terms with additional paragraphs and/or sections devoted to the impact(s) of sex on disease onset, progression, and importantly healthcare, along with potential disparities.
2. ApoE impact. The ApoE4 allele is a major risk factor for AD and other neurodegenerative proteinopathies- and is not mentioned in this review. The authors are urged to add ‘ApoE’ to their search terms and possibly generate a paragraph or so on how this may impact healthcare of affected individuals, their management teams, and caregivers.
3. ‘Omics studies. ‘Omics (e.g., transcriptomics, proteomics, metabolomics, among others) are not mentioned in the scoping review. Additional discourse on how these high throughput approaches may impact diagnosis, treatment, and overall healthcare providing is recommended.
Minor point: Disparities are not given much attention, such as socioeconomic status, racial/ethnic bias, or rural location, among others (e.g., pollution, access to healthcare systems) and are worthy of additional discourse.
Round 2
Reviewer 2 Report
Comments and Suggestions for Authors
The authors addressed all my comments. I have no further comments.
Reviewer 3 Report
Comments and Suggestions for Authors
The authors have thoroughly addressed my comments and incorporated the necessary revisions. The paper now presents significantly greater substantive and scientific value.